

# Some Rigorous Results on Symmetry Breakings in Gauge QFT

**Franco Strocchi**

Dipartimento di Fisica, Università di Pisa, Pisa, Italy

## Abstract

The extraordinary success of the Standard Model asks for a more rigorous control beyond the perturbative approach, which is affected by mathematical problems (interaction picture and canonical quantization, non-covergence of the perturbative series, triviality results for $\phi^4$ model and related models, etc.). We shall briefly discuss some non-perturbative results concerning the crucial role and realization of symmetry breakings in the Standard Model.

## 1 BEH mechanism

A crucial structural ingredient of the standard model is the BEH (briefly Higgs) mechanism, which has been discussed and used at the perturbative level with an expansion based on a mean field ansatz. Since mean field expansions are known to give incorrect results for the critical temperature and the energy spectrum in spin models, one would like to control the mechanism with a rigorous non-perturbative approach. This will also free the conclusions from the weak points of the perturbative expansion, which is known to lead to a non-convergent series and relies on the interaction picture and canonical quantization, both mathematically excluded for non-trivial interactions [1].

As it is well known, but sometime overlooked in the textbook discussions of the Higgs mechanism, in order to avoid the exclusion of a symmetry breaking order parameter by Elitzur theorem, the first crucial step it to introduce a gauge fixing; then, the discussion of the way the mechanism avoids the occurrence of massless Goldstone bosons becomes gauge fixing dependent. Furthermore, in view of a non-pertubative approach, a mean field term, which breaks the global gauge group, should not appear in the gauge fixing, because it would require an *a priori* non-perturbative control of its selfconsistency. One then considers gauge fixings invariant under the global gauge group.[1]

---

[1]The motivations for such strategic choices, which exclude the pathological unitary gauge as well as the so-called $\xi$ gauges, are discussed in Ref. [1]

From a rigorous point if view, one faces the problem of proving that the spontaneous breaking of the global gauge group does not imply the existence of massless Goldstone bosons. This shall be dealt with in the BRST gauge and, in the abelian case, in the Coulomb gauge.

i) *Absence of Goldstone particles in the Higgs mechanism*

We choose to discuss the problem in the BRST gauge. The advantage is that the corresponding field algebra $\mathcal{F}$ is local, so that one may control the generation of the infinitesimal transformations, under the global gauge group, by conserved local currrents, $J_\mu^a$, as in the proof of the Goldstone Theorem, thanks to the non-renormalization theorem for the commutators of local conserved currents with local fields:

$$< \delta^a F >= i \lim_{R\to\infty} < [Q_R^a, F] >, \quad \forall F \in \mathcal{F}, \tag{1.1}$$

where $Q_R^a$ denotes the suitably regularized charge localized in the sphere of radius $R$

$$Q_R^a = J_0^a(f_R \alpha) = \int d^4x\, J_0^a(x) f_R(\mathbf{x})\, \alpha(x_0), \tag{1.2}$$

$f_R(\mathbf{x}) = f(|\mathbf{x}|/R)$, $f(x) = 1$, for $|x| \le 1$, $f(x) = 0$, for $|x| > 1 + \varepsilon$, $f, \alpha$ infinitely differentiable, supp $\alpha \subseteq [-\varepsilon, \varepsilon]$, $\tilde{\alpha}(0) = \int dx_0\, \alpha(x_0) = 1$.

**Theorem 1.1** *(Higgs mechanism) In the BRST quantization of a Yang-Mills theory, the spontaneous breaking of a one-parameter subgroup of the global gauge group G by the vacuum expectation of $F \in \mathcal{F}$, $< \delta^a F > \ne 0$, implies the existence of a $\delta(k^2)$ singularity in the Fourier transform of $< F J_\mu^a(x) >$, (**massless Goldstone modes** in the a-channel), where $J_\mu^a(x)$ is the conserved current which generates the infinitesimal transformations of the local fields under such a one-parameter subgroup; however, such modes cannot describe physical particles.*

The basic ingredient for the proof is the validity of the Local Gauss Law, on the physical states, which in the BRST gauge reads

$$J_\mu^a = \partial^\nu F_{\mu\nu}^a + \{ Q_B, (D_\mu \bar{c})^a \}, \tag{1.3}$$

where $Q_B$ denotes the BRST charge and $\bar{c}$ one of the ghost fields [1, 2]. The physical state vectors $\Psi$ are selected by the BRST subsidiary condition

$$Q_B \Psi = 0, \quad \Rightarrow \quad < \Psi, (J_\mu^a - \partial^\nu F_{\mu\nu}^a)\Psi > = 0. \tag{1.4}$$

One may prove that this excludes a contribution to such a massless mode by a physical state, as intermediate state in the two-point function $< F J_\mu^a(x) >$ [3, 4]. The unphysical nature of the massless modes in local renormalizable gauges has been argued within a perturbative expansion (see the very comprehensive review [5]). The above non perturbative result improves the perturbative analysis, since it does not rely on a semiclassical mean field ansatz nor on the summability of the perturbative series; moreover the order parameter is not restricted to be a pointlike field.

ii) *A theorem on the abelian Higgs mechanism*

In the abelian case one may prove a sharper result [1, 6, 7], which includes a rigorous link between the disappearance of the Goldstone boson and the vector boson becoming massive (beyond the popular anthropomorphic picture of the first being eaten by the latter).

This is obtained in the Coulomb gauge; the advantage is that the corresponding field algebra $\mathcal{F}_C$ does not contain unphysical fields and all the states of its Hilbert space representation have a physical meaning. On the other hand, the validity of the Local Gauss Law implies that

$\mathcal{F}_C$ is non-local; as a consequence, one looses the control of the local generation of the infinitesimal transformations $\delta^{U(1)}$ of the fields under the global $U(1)$ gauge group $\beta^\lambda, \lambda \in \mathbf{R}$:

$$\delta^{U(1)}F \equiv \frac{d\beta^\lambda(F)}{d\lambda}|_{\lambda=0}, \; F \in \mathcal{F}_C. \tag{1.5}$$

Since the charged fields $F$, characterized by $\delta^{U(1)}F \neq 0$, are non-local, the non-renormalization theorem for the commutators of conserved currents does not apply, and in fact, contrary what is usually taken for granted, one may prove that the commutatators of the space integral of the current charge density $j_0(\mathbf{x}, x_0)$ are not independent of the time $x_0$

$$\lim_{R\to\infty} [\, j_0(f_R, x_0), \varphi^C(y)\,] = -e \int dm^2 \rho(m^2) \cos(m(x_0 - y_0)) \varphi^C(y), \tag{1.6}$$

where $\varphi^C$ is the Coulomb charged field and $\rho(m^2)$ is the spectral measure which defines the two point function of the vector boson field $F_{\mu\nu}$

$$< F_{\mu\nu}(x)F_{\lambda\sigma}(y) >= id_{\mu\nu\lambda\sigma} \int dm^2 \rho(m^2) \Delta^+(x - y; m^2), \tag{1.7}$$

$$d_{\mu\nu\lambda\sigma} = g_{\nu\sigma}\partial_\mu\partial_\lambda + g_{\mu\lambda}\partial_\nu\partial_\sigma - g_{\nu\lambda}\partial_\mu\partial_\sigma - g_{\mu\sigma}\partial_\nu\partial_\lambda.$$

In order to find a relation between the current charge density $j_0(f_R, x_0)$ and the electric charge, at least in the unbroken case, an improved smearing is needed [8,9] which amounts to introducing $Q_{\delta R} \equiv j_0(f_R \alpha_{\delta R})$, $\alpha_{\delta R} \equiv \alpha(x_0/\delta R)/\delta R$, $0 < \delta < 1$ ($\alpha$ as in eq. (1.2)). Then, the so obtained charge has time independent commutators and annihilates the vacuum $\Psi_0$

$$\delta_c F \equiv i \lim_{\delta\to 0, R\to\infty} [\, Q_{\delta R}, F\,], \quad \lim_{R\to\infty} Q_{\delta R}\Psi_0 = 0. \tag{1.8}$$

The next question is the relation between the derivation $\delta_c$, induced by the current charge density and the derivation $\delta^{U(1)}$. Such a relation turns out to play a crucial role for the following general theorem on the Higgs phenomenon.

**Theorem 1.2** *(Higgs phenomenon)*
**A.** *The current and the $U(1)$ derivations coincide, $\delta_c = \delta^{U(1)}$, if and only if the two point spectral measure of the vector field $F_{\mu\nu}$ contains a $\delta(m^2)$, namely if the corresponding **vector boson is massless**; in this case, the global $U(1)$ **is unbroken** and the matrix elements of its generator $Q$ are given by*

$$< \Psi, Q\Phi >= \lim_{\delta\to 0, R\to\infty} < \Psi, Q_{\delta R}\Phi >, \tag{1.9}$$

*(for all the Coulomb states $\Psi, \Phi$). Thus, thanks to the improved smearing, one recovers the expected **relation between the charge density and the $U(1)$ charge**, although in an alerted form.*

**B.** *The global $U(1)$ **gauge group is broken**, i.e. $< \delta^{U(1)} F >\neq 0$, $F \in \mathcal{F}_C$, only if $\delta^{U(1)} \neq \delta_c$ (i.e. the relation between current and infinitesimal transformations is lost) and in this case,*
*i) the **vector boson is massive**;*
*ii) the Goldstone spectrum, defined by the Fourier transform of the two point function $< j_0(x) F >$, is governed by the spectral function of the vector field, and therefore cannot contain any $\delta(k^2)$ (i.e. there are **no associated Goldstone bosons**);*
*iii) the Gauss charge, defined by the suitably smeared flux of $F_{0i}$ at space infinity, vanishes on the Coulomb states (**screening of the Gauss charge**):*

$$\lim_{\delta\to 0, R\to\infty} < \Psi, Q_{\delta R}\Phi >= 0. \tag{1.10}$$

## 2 Gauge group topology solves the $U(1)$ problem and yields the $\theta$ vacuum structure

A corner stone for the control of QCD structure is the discovery of the role of topology, yielding the $\theta$ vacuum structure and a solution of the $U(1)$ problem. The standard treatment is based on the assumption that the euclidean functional integral is governed by the instanton solutions (semiclassical approximation), which are classified by their topological winding number $n$. The functional integral is thus evaluated by first integrating over the class of euclidean (continuous) configurations with given *winding number $n$* and then by summing over $n$, with a weighting factor $e^{i\theta n}$, where $\theta$ is a *free* parameter, the so-called $\theta$ *angle*.

Such a procedure is not free of mathematical problems, since, already in the free field case, the set of continuous euclidean configurations has zero functional measure. Hence, in contrast with the quantum mechanical case, the WKB (semiclasssical) approximation is problematic in QFT.

Furthermore, such an approach does not clearly settle the debated question of whether the axial $U(1)_A$ transformations may be still defined for the observable fields, so that $U(1)_A$ is *spontaneously* broken in QCD.

A rigorous solution of such problems may be obtained following a suggestion by Roman Jackiw [4, 10]. The idea is to directly exploit the non-trivial topology of the gauge group, rather than its reflexes on the classification of the instanton solutions; in this way one does not make any reference to the problematic semiclassical instanton approximation.

ii) *Solution of the $U(1)$ problem*

This is conveniently obtained in the temporal gauge, which is local and positive; the only delicate (but mathematically crucial) point is that, as a consequence of the required invariance of the vacuum under the Local Gauss Law operator, the represented field algebra $\mathcal{F}$ is generated by the gauge invariant fields and by the formal exponentials of the gauge dependent fields (with algebraic relations corresponding to those of their formal exponentials). [1, 4, 10]

The first step is the definition of time independent $U(1)_A$ transformations $\beta^\lambda, \lambda \in \mathbf{R}$ of $\mathcal{F}$ and, in particular, of its observable subalgebra $\mathcal{F}_{obs}$

$$\beta^\lambda(F) = \lim_{R\to\infty} V_R^5(\lambda) F V_R^5(-\lambda), \quad \forall F \in \mathcal{F}, \tag{2.1}$$

where the one-parameter unitary operators $V_R^5(\lambda)$ are (formally) the exponentials $e^{i\lambda J_0^5(f_R\alpha)}$, with $J_\mu^5$ the conserved (gauge dependent) current

$$J_\mu^5 = j_\mu^5 - (16\pi^2)^{-1}\varepsilon_{\mu\nu\rho\sigma}\mathrm{Tr}\left[F^{\nu\rho}A^\sigma - (2/3)A^\nu A^\rho A^\sigma\right] \equiv j_\mu^5 + K_\mu^5, \tag{2.2}$$

( $j_\mu^5$ is the gauge invariant anomalous current); by locality, the limit is reached for finite $R$.

The gauge dependence of the unitary operators $V_R^5(\lambda)$ does not invalidate the above definition, since they are merely instrumental for the definition of the chiral transformations $\beta^\lambda$ on the observable fields, a result which is clearly independent of the gauge fixing and of the corresponding (gauge dependent) field algebra in which $\mathcal{F}_{obs}$ is embedded. It looks short sighted [11] to blame on the fact that $J_\mu^5$ or the (better behaved) exponentials $V_R^5(\lambda)$ are gauge dependent not observable operators; such a point of view would in fact deny the very existence of the non-abelian gauge symmetries of the standard model, being generated by gauge dependent currents.

Given the *existence of the $U(1)_A$ transformations of the observable fields*, a gauge independent fact, no matter how its actual existence is proved, the real issue is the mechanism for evading the Goldstone theorem, for which the non-abelianess of the gauge group should play a decisive role.

The next step is the interplay between axial transformations and gauge transformations. To this purpose we analyse the properties of the local gauge group $\mathcal{G}$, left unbroken by the gauge fixing, with elements $\alpha_{\mathcal{U}}$ parametrized by time independent $C^{\infty}$ unitary functions $\mathcal{U}(\mathbf{x})$, taking value in the global group $\mathbb{G}$ and differing from the identity only on a compact set, $\mathcal{K}_{\mathcal{U}} \subset \mathbf{R}^3$. Thanks to their space localization the $\mathcal{U}$ obviously extend to the one-point compactification of $\mathbf{R}^3$, $\dot{\mathbf{R}}^3$, which is isomorphic to the three sphere $S^3$, and define *continuous* mappings of $S^3$ onto the global gauge group $\mathbb{G}$: $\mathcal{U}(\mathbf{x}) : \dot{\mathbf{R}}^3 \sim S^3 \to \mathbb{G}$. Such mappings $\mathcal{U}$ fall into disjoint homotopy classes labeled by the (topological invariant) **winding number** $n(\mathcal{U})$

$$n(\mathcal{U}) = (24\pi^2)^{-1} \int d^3x \, \varepsilon^{ijk} \operatorname{Tr}[\mathcal{U}_{(i)}(\mathbf{x})\mathcal{U}_{(j)}(\mathbf{x})\mathcal{U}_{(k)}(\mathbf{x})] \equiv \int d^3x \, n_{\mathcal{U}}(\mathbf{x}), \qquad (2.3)$$

$\mathcal{U}_{(i)}(\mathbf{x}) \equiv \mathcal{U}(\mathbf{x})^{-1}\partial_i \mathcal{U}(\mathbf{x})$; $\mathcal{U}_n$ shall denote a function with winding number $n$.

The one-parameter groups of unitary gauge functions

$$\mathcal{U}(\lambda g) = e^{i\lambda g(\mathbf{x})}, \quad \lambda \in \mathbf{R}, \quad g(\mathbf{x}) = g_a(\mathbf{x}) T^a, \quad g_a \in \mathcal{D}(\mathbf{R}^3),$$

($T^a$ the representative matrices of the generators of $\mathbb{G}$) continuously connected to the identity, define a subgroup $\mathcal{G}_0 \subset \mathcal{G}$, which is generated by the unitary operators $V(\mathcal{U}(\lambda g)) \in \mathcal{F}$, formally the exponentials of the Gauss operator $G^a \equiv (\mathbf{D} \cdot \mathbf{E})^a - j_0^a$, $j_{\mu}^a = i\bar{\psi}\gamma_{\mu}t^a\psi$, $V(\mathcal{U}(\lambda g)) \sim e^{i\lambda G(g)}$, $G(g) \equiv \sum_a G^a(g_a)$, $g_a \in \mathcal{D}(\mathbf{R}^3)$, since formally $\delta^{g_a}F = i[G^a(g^a), F]$, (the operators $V(\mathcal{U}(\lambda g))$ need not to be represented by weakly continuous unitary operators). $\mathcal{G}_0$ is called the *Gauss subgroup* of $\mathcal{G}$ and its elements have zero winding number. In the following, for simplicity, we shall often adopt the short-hand notation $\mathcal{U}(g)$, or $\mathcal{U}_g$.

In the Hilbert space $\mathcal{H}$ defined by the correlation functions of a vacuum state $\omega_0$, the physical state vectors $\Psi$ are selected by the subsidiary condition

$$V(\mathcal{U}(\lambda g))\Psi = \Psi, \quad \forall \, \mathcal{U}(\lambda g) \in \mathcal{G}_0, \quad \Psi \in \mathcal{H}' \subset \mathcal{H}. \qquad (2.4)$$

By exploiting the localization property of the gauge functions and the locality of $\mathcal{F}$, one shows that the state $\omega_0$ is invariant under the full group $\mathcal{G}$, namely $\omega_0(\alpha_{\mathcal{U}}(F)) = \omega_0(F)$, $\forall F \in \mathcal{F}$, so that $\mathcal{G}$ is implemented by unitary operators $V(\mathcal{U})$ in $\mathcal{H}$ and

$$V(\mathcal{U}_n) V_R^5(\lambda) V(\mathcal{U}_n)^{-1} = e^{i\lambda \, 2n} V_R^5(\lambda). \qquad (2.5)$$

**Proposition 2.1** *The spontaneous breaking of the $U(1)_A$ symmetry $\beta^{\lambda}$ in QCD, by $< \delta^5 A > \neq 0$, with $\delta^5 A$ the infinitesimal $U(1)_A$ transformation of $A$ and $A$ an observable (hermitian) field, evades the Goldstone theorem because $\delta^5 A$ cannot be related to the two point function of $A$ and a (local conserved) current, as required for the proof of the Goldstone theorem.*

The point is that, as a consequence of the non-trivial topology of $\mathcal{G}$ and of the above equations, one has

$$< V_R^5(\lambda)A > = < \alpha_{\mathcal{U}_n}(V_R^5(\lambda)A) > = e^{i2n\lambda} < V_R^5(\lambda)A > .$$

This proves that $< V_R^5(\lambda)A >$ is a singular function of $\lambda$ and its derivative with respect to $\lambda$ does not exists; then, even if the axial $U(1)_A$ transformations are given by the action of the local unitary operators $V_R^5(\lambda)$, their infinitesimal form cannot be given by commutators with a local conserved current, here $J_{\mu}^5$, a crucial assumption for the proof of the Goldstone theorem.

ii) *Gauge topological group and $\theta$ vacuum structure*

The topology of the gauge group $\mathcal{G}$ is described by the quotient of $\mathcal{G}$ by its normal subgroup $\mathcal{G}_0$. $\mathcal{T} \equiv \mathcal{G}/\mathcal{G}_0$, is an abelian group with elements $\mathcal{T}_n$ which are classified by the (topological) winding number $n$, and commute with the gauge transformations.

**Theorem 2.2** *In the Hilbert space representation of the temporal gauge field algebra $\mathcal{F}$, by a (Gauss invariant) vacuum state $\omega_0$, the topological group $\mathcal{T}$ is represented by gauge invariant operators $T_n$, which belong to the center of the algebra of observables $\mathcal{A}$ and reduce to unitary operators on the space $\mathcal{H}'$ of physical states, with spectrum $e^{i\,2n\,\theta}$, $\theta \in [0, \pi)$. Thus, the (factorial) representations of the observable algebra in the physical space are labelled by the angle $\theta$.*

*Under $U(1)_A$ transformations $\beta^\lambda$, one has*

$$\beta^\lambda(T_n) = e^{i\,2n\,\lambda}\, T_n, \tag{2.6}$$

*so that in each representation of $\mathcal{A}$ with trivial center, $U(1)_A$ is **always** broken.*

*If the Gauss invariant vacuum state $\omega_0$ defines an irreducible representation of the field algebra $\mathcal{F}$, then it selects a definite value of $\theta$ ($\theta$ **vacuum structure**)*

$$T_n \Psi_0 = e^{i\,2n\,\theta}\, \Psi_0. \tag{2.7}$$

Thus, the $\theta$ angle arises in an intrinsic way as a label of the spectrum of the center of the algebra of observables, uniquely selected in each irreducible (vacuum) representation of the field algebra, rather than as a free parameter in the semiclassical instanton approximation of the euclidean functional integral [4].

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
