# Peer review of "Some rigorous results on symmetry breakings in gauge QFT"

_SciPost Physics Proceedings, doi:SciPost Phys. Proc. 6, 021 (2022)_

## Round 1 · Referee Report · Anonymous (Referee 1) · 2022-2-25

Report
After outstanding successes of the SM predictions, especially the Higgs boson,
the formal foundations of the theory are again on the agenda.
These concern , e.g. the mathematical consistency of the scalar sector of the SM, development of the non-perturbative methods etc. In the review by F. Strocchi both earlier and recent results on these issues are presented in an excellent manner which allow even to non-experts to catch the main points.
the formal foundations of the theory are again on the agenda.
These concern , e.g. the mathematical consistency of the scalar sector of the SM, development of the non-perturbative methods etc. In the review by F. Strocchi both earlier and recent results on these issues are presented in an excellent manner which allow even to non-experts to catch the main points.

---

## Round 1 · Referee Report · Anonymous (Referee 2) · 2022-3-28

Strengths
Very important and difficult problems concerning foundations of the standard model are treated: the Higgs mechanism at the quantum level in the BRST gauge beyond perturbation theory and the arguments for the $\theta$ vacuum in QCD beyond semiclassical approximation
Weaknesses
No comments on taking Gribov copies into consideration are made
Report
The importance of this study is to fill in two gaps in the foundations of the standard model and to clarify the related issues traditionaly missing from textbooks on quantum field theory. The former gap concerns formulation of the Higgs mechanism at the quantum level and beyond perturbation theory. It is shown that, in the BRST gauge in non-Abelian gauge theory, the Goldstone bosons are associated with nonphysical excitations. The latter gap is related to the so called theta vacuum in QCD, the existence of which in the framework of the conventional instanton approach is only hypothetical, although plausible. Without resort to instantons, the author found that the condition under which the ground state is characterized by a definite value of $\theta$. This condition is related to spontaneous breaking of the axial $U_A(1)$ symmetry (Gauss-invariant vacuum state defines an irredusible representation of the temporal-gauge field algebra); this being so, some conditions of the Goldstone theorem are not fulfilled.
Worth publishing in the SciPost Physics Proceedings
Worth publishing in the SciPost Physics Proceedings
Requested changes
It is advisable to use the SciPost templates

---

## Round 1 · Referee Report · Anonymous (Referee 3) · 2022-3-29

Strengths
1 - Thorough, rigorous and detailed presentation of the Higgs mechanism in the nonabelian
gauge field theory.
gauge field theory.
Report
Two topics are addressed. As a first topic the Higgs mechanism is considered. The nonperturbative prove of the unphysical nature of the massless modes in local renormalizable gauges is recalled.
The second topic is the U(1) problem and $\theta$-vacua. It is demionstrated that within the approach used by the author the $\theta$ angle arises in an intrinsic way rather than as a free parameter in
the semiclassical instanton approximation.
I recommend the manuscriopt for publication.
The second topic is the U(1) problem and $\theta$-vacua. It is demionstrated that within the approach used by the author the $\theta$ angle arises in an intrinsic way rather than as a free parameter in
the semiclassical instanton approximation.
I recommend the manuscriopt for publication.
Requested changes
The original scipost template should be used

---

## Editorial Decision

published